# CPX-351: An Old Scheme with a New Formulation in the Treatment of High-Risk AML

**DOI:** 10.3390/cancers14122843

**Published:** 2022-06-08

**Authors:** Matteo Molica, Salvatore Perrone, Carla Mazzone, Laura Cesini, Martina Canichella, Paolo de Fabritiis

**Affiliations:** 1Hematology Unit, S. Eugenio Hospital, ASL Roma 2, 00144 Rome, Italy; cmazzone1@virgilio.it (C.M.); laura.cesini@aslroma2.it (L.C.); martina.canichella@aslroma2.it (M.C.); paolo.de.fabritiis@uniroma2.it (P.d.F.); 2Hematology, Polo Universitario Pontino, S.M. Goretti Hospital, 04100 Latina, Italy; s.perrone@ausl.latina.it; 3Department of Biomedicina and Prevenzione, Tor Vergata University, 00133 Rome, Italy

**Keywords:** acute myeloid leukemia, therapy-related acute myeloid leukemia, acute myeloid leukemia with myelodysplasia-related changes, CPX-351

## Abstract

**Simple Summary:**

Secondary AML (s-AML) including therapy-related acute myeloid leukemia (t-AML) and acute myeloid leukemia with myelodysplasia-related changes (AML-MRC) represent approximately one quarter of all AML cases. These AML subcategories are predominantly associated with advanced age and present a specific biologic profile including adverse genetics and a multidrug resistance phenotype, which often determine dramatically poor outcomes after conventional chemotherapy. In 2017, the FDA approved CPX-351, a liposomal formulation of cytarabine and daunorubicin at a fixed 5:1 molar ratio, for the treatment of adults with newly diagnosed t-AML and MRC-AML. Since the approval, many trials have been conducted or are still ongoing in order to assess the role of CPX-351 as frontline treatment in different AML subcategories, as a potential bridge to transplant or in combination with target therapies. In this review, we will discuss the current role of CPX-351 in treating these high-risk AML, focusing on how its use may potentially change the treatment paradigms of AML.

**Abstract:**

Therapy-related acute myeloid leukemia (t-AML) and acute myeloid leukemia with myelodysplasia-related changes (AML-MRC) represent aggressive diseases characterized by a dismal prognosis if compared with de novo acute myeloid leukemia, especially in older patients. In these AML subsets, standard chemotherapy regimens produce poor response rates and unsatisfactory outcomes. Historically, conventional approaches consisted of an anthracycline combined with continuous infusion of cytarabine for 7 days, the “3+7” regimen. Several attempts have been conducted to ameliorate this combination regimen but inconsistent improvements in response rates and no significant changes in overall survival have been observed, until the recent introduction of targeted molecules. A liposomal formulation of traditional chemotherapy agents cytarabine and daunorubicin, termed CPX-351, enhances pharmacodynamics and synergistic effects through the maintenance of the optimal 5:1 molar ratio, which extends the treatment’s half-life and increases the bone marrow tropism of the drug. The use of CPX-351 in newly diagnosed AML-MRC and t-AML patients aged 60–75 years has demonstrated superior remission rates compared to conventional chemotherapy and improvements in event-free and overall survival. Recently, published data from a 5-year follow-up highlighted evidence that CPX-351 has the ability to produce and contribute to long-term remission and survival in older patients with newly diagnosed high-risk/secondary AML. Future perspectives include evaluation of dose intensification with CPX-351 in high-risk settings, combining this agent with targeted therapies, and better understanding the mechanism of improved responses in t-AML and AML-MRC. In this review, we will examine the role of CPX-351 inside the new AML therapeutic scenario and how its employment could potentially modify the treatment algorithm of high-risk and elderly patients with AML

## 1. Introduction

Acute myeloid leukemia (AML) is a hematopoietic progenitor cell clonal disease in which a population of leukemic stem cells drives the proliferation of aberrant myeloid precursor cells (blasts) that become unable to differentiate [1]. AML is the most common type of leukemia, representing approximately 25% of all leukemia in adults in the Western world. 

The incidence of AML rises proportionally with age, from 1.8 cases per 100,000 people below the age of 65 to 13.7 instances per 100,000 people over the age of 65. In developed countries, more than half of newly diagnosed AML patients are over 65 years old, with a median age at the time of diagnosis of 67, and AML more frequently occurs in men than in women. The prevalence of AML in the European Union is predicted to be 1.1 per 10,000 people [2].

From a biological perspective AML is considered a heterogeneous disease that can be classified in three different subgroups based on the clinical ontogeny.

De novo AML occurs without a previously documented exposure to potential leukemogenic treatments or prior hematologic disorder such as myelodysplastic syndrome (MDS).Therapy-related AML (t-AML) occurs as a postponed complication in patients who previously received leukemogenic treatments.Secondary AML (s-AML) consists of a third group of AML involving both patients who have a history of prior chemotherapy/radiotherapy (t- AML) or AML occurring consequently to a previous hematologic disease such as MDS, myeloproliferative disorders, combined myelodysplasia, and myeloproliferative disease, or chronic myelomonocytic leukemia (CMML). This third subgroup represents approximately 10–30% of all AML cases and often presents lower response rates, shorter duration of remission, and a worse overall survival (OS) compared with de novo AML [3].

For several decades, patients with newly diagnosed AML have received as a gold standard induction regimen the therapeutic combination of cytarabine, a nucleoside analog, with an anthracycline, most frequently daunorubicin or idarubicin [4]. This treatment combination, called “3+7”, continues to be the standard approach for more than 40 years although it determines poor rates of complete response (CR) in elderly AML cases presenting specific AML genomic and cytogenetic risk characteristics frequently associated with a dismal prognosis. Therefore, outcomes continue to be unsatisfactory in older AML (>70 years) and cases with s-AML [5]. While many attempts aimed to ameliorate outcomes after the induction approach by adding other agents or intensifying post-remission treatment did not lead to significant results, superior response rates and higher OS have recently been produced by CPX-351, a liposomal encapsulation of cytarabine and daunorubicin at a fixed 5:1 synergistic molar ratio [6,7]. Data from a phase-III trial that explored the use of CPX-351 in 309 patients with s-AML or t-AML aged 60–75 years, determined the approval of this combined agent by FDA and EMA in 2017 and 2018, respectively, for adult patients with newly diagnosed s-AML or t-AML.

In this review, we will examine the role of CPX-351 inside the new AML therapeutic scenario and how its employment could potentially modify the treatment algorithm of high-risk and elderly patients with AML.

## 2. Secondary (s-AML) and Therapy-Related (t-AML) Acute Myeloid Leukemia

Secondary AML identifies two different leukemic evolutions: (1) developing from antecedent myeloproliferative disorder (MPN), myelodysplasia (MDS), or aplastic anemia (AA) with or without treatment; and (2) as a consequence of prior exposure to a documented leukemogenic agent recognized as therapy-related AML (t-AML).

t-AML definition includes both a process associating with the nature of its neoplastic onsets and holds a negative connotation, correlating with a poor prognosis [8,9,10]. s-AML, although often used interchangeably with t-AML, is a wider and more specific definition, which additionally includes AML anteceded by a hematological disorder regardless of treatments administered during the first disease. In the 2008 revision, the World Health Organization (WHO) redefined the AML classification paradigm by including “AML with myelodysplasia-related changes (MRC-AML)” that involves AML that occurred from prior MDS/MPN, AML with MDS related cytogenetic abnormality and AML with morphologic multi-lineage dysplasia [11]. In the WHO’s 2016 updated revision, AML-myelodysplasia-related changes and “therapy related myeloid neoplasms” have been well-defined as specific subcategories of AML [1]. Lindsley and colleagues used mutational pattern profiles to divide clinically diagnosed t-AML into several genetic ontogeny subgroups [12]. Attempts by the same group of researchers to resolve clinical-pathologic heterogeneity within AML revealed a core collection of mutations in selected genes that were highly specific for s-AML. A mutation in *SRSF2*, *SF3B1*, *U2AF1*, *ZRSR2*, *ASXL1*, *EZH2*, *BCOR*, or *STAG2* was found to be >95% specific for s-AML diagnosis. These mutations appear early in leukemogenesis and frequently remain in clonal remissions, according to an analysis of serial samples from individual patients.

In t-AML and elderly de novo AML patients, these mutations identify a specific genetic subtype, which shows similar clinical-pathologic properties with s-AML and recognizes a subgroup of patients with dismal outcomes, characterized by a lower CR rate, more frequent re-induction, and reduced event-free survival (EFS) [12]. Thus, patients with t-AML more commonly present abnormal cytogenetic characteristics including an augmented prevalence of adverse-risk karyotypes [13]. According to the chemotherapeutic agent and/or radiation previously received, two subsets of t-AML can be recognized. The most frequent subgroup, arising after exposure to alkylating drugs and/or radiation with a latency period of 5–10 years, is predominantly characterized by unbalanced cytogenetic alterations, such as loss or deletion of chromosomes 5 and/or 7 [14]. The second less frequent subgroup, onsetting after therapies including agents targeting topoisomerase II, has shorter latency period of 1–5 years and often harbors balanced chromosomal abnormalities, which involve *MLL*, *RUNX1*, and *PML-RARA* genes [15]. However, in recent years most patients have been treated with both alkylating agents and agents that target topoisomerase II for prior tumors; therefore, a distinct differentiation based on the type of previous treatment is often impossible. Furthermore, mutations of the *TP53* tumor suppressor gene have also been widely observed in t-AML. These alterations have been detected in less than 10% of patients with de novo AML, while they are found in more than 20% and 90% of cases of t-AML and erythroid leukemia, respectively [16]. *TP53* aberrations are characterized by gene mutations, mainly located in the DNA binding domain of the gene, and/or deletions of several sizes involving the *TP53* locus on chromosome 17p13. Although most *TP53* mutations result as somatically acquired and substantially constitute an early leukemogenic predisposition, *TP53* germline mutations are increasingly being known, especially in t-AML patients [17]. *TP53* aberrations correlated with an extremely adverse prognosis as reported on different independent studies [18]. Different factors may be correlated with the poor outcomes observed in t-AML patients [19,20] and may determine worse outcomes in s-AML [19,20]. Patients may be older and may present different organ dysfunction derived from potential comorbidities. Furthermore, these patients may have a long-term malignancy or a relapse of their underlying cancer. Prior treatment or MDS can deplete hematopoietic reserves, prolonging myelosuppression after AML treatment and predisposing patients to more severe treatment-related complications. Finally, molecular mutations and cytogenetic abnormalities such as *TP53* mutations may make conventional chemotherapy ineffective.

## 3. CPX-351 Mechanism of Action

CPX-351 is an example of the CombiPlex platform (Celator Pharmaceuticals) that, conversely to standard combination chemotherapy approaches, recognizes synergistic drug ratios in vitro and operates a suitable nanoscale carrier to increase drug delivery [21]. CPX-351 maintains a 5:1 molar ratio of cytarabine:daunorubicin co-encapsulated within a bi-lamellar liposome, enabling intracellular delivery of the synergistic drug ratio and enhancing uptake in leukemia cells to a greater extent than normal cells. Cytarabine is a classic nucleotide-analogue chemotherapeutic drug first used in 1969. Cytarabine enters into cells mainly via nucleoside transporters, including SLC29A1 and is subsequently phosphorylated by several nucleoside kinases into its active form cytarabine triphosphate (Ara-CTP). Ara-CTP competes with deoxycytidine triphosphate (dCTP) for incorporation into DNA, interferes with DNA and RNA synthesis by polymerases, and ultimately causes cell death [22].

The second drug, daunorubicin, is an anthracycline antibiotic introduced in the 1970s as part of the standard “3+7” regimen for AML [4]. Anthracyclines intercalate with DNA forming heterotrimeric complexes with topoisomerase II and DNA to directly inhibit transcription and replication. They create radical intermediates that react with O_2_ to produce superoxide anion radicals such as H_2_O_2_ and •OH that oxidize DNA bases, leading to apoptosis [23].

The pharmacokinetics of the two drugs are very different: cytarabine peak concentrations of 2–50 μM are measurable in plasma after intravenous injection of 30–300 mg/m^2^ but fall rapidly (t1/2 ≈ 10 min) and less than 10% of the injected dose is excreted unchanged in the urine within 12–24 h [24]. In contrast, the plasma disappearance curve for daunorubicin is multiphasic, with a terminal t1/2 of 30 h. Therefore, by itself, cytarabine is bio-available only briefly and continuous infusion is needed. CPX-351 provides an elegant solution to this problem because the two drugs are co-encapsulated in the same liposome; this combination mitigates the pharmacokinetic disparity between the two drugs because the liposomes determine the distribution of both drugs within the body and the preferential delivery of them to the malignant blasts [25]. In fact, CPX-351 prolongs the estimated mean t1/2 of cytarabine and daunorubicin to 40 and 31 h, respectively [26]. Tissue biodistribution of CPX-351 is different from the free drugs: concentrations are higher in well-perfused organs and tissues, such as bone marrow, spleen, liver, kidney, and lymph nodes and lower in less perfused tissues, such as skin or fat [25]. As observed in murine models [21], this translates into clinical implications because tissues where AML is present such as bone marrow and the reticuloendothelial system will be more exposed to the drug, while toxicity such as alopecia and gut dysfunction might be reduced [25]. Using CPX-351, it is also feasible to administer in vivo an approximate 5:1 molar ratio of cytarabine and daunorubicin, in vitro that ratio demonstrated to be maximally synergistic with minimal antagonistic anti-leukemia activity [26].

## 4. Clinical Trials

### 4.1. Phase I Trials

A first in-human, dose-escalation, phase I clinical study was initiated in 2011 to assess the pharmacokinetic profile, maximum tolerated dose (MTD), and toxicity/safety of CPX-351 in patients with relapsed or refractory (R/R) AML, or acute lymphoblastic leukemia (ALL) and high-risk MDS. Participants were characterized by an adequate organ function, a valid left ventricular ejection fraction (LVEF) (>50%), and satisfactory Eastern Cooperative Oncology Group (ECOG) performance status (≤2) [27]. Daunorubicin and cytarabine liposome were administered at escalating doses from 3 to 134 U/m^2^; all patients had been treated with a median of two previous treatments, and CPX-351 represented a first salvage approach for 23 AML patients. In single-patient cohorts, the first cycle of CPX-351 was administered at 3 U/m^2^ with increasing dosages until treatment-related toxicities or a significant reduction in bone marrow cellularity or blast count were noted.

Subsequent cohorts included 3 patients and CPX-351 doses were increased by 33% until the occurrence of dose-limiting toxicity (DLT). Cohorts were expanded only in case of the occurrence of an adverse event or if patients were not evaluable. Bone marrow was assessed on day 14 with a goal of achieving more than 20% reduction with less than 5% blasts. After the investigators’ consent, a second induction cycle on days 1 and 3 was allowed. Patients who had obtained CR were considered eligible for a further consolidation cycle on days 1 and 3. In order to add further data on safety and preliminary efficacy of the drug, an expansion cohort of patients with AML in first relapse was included at the MTD [28]. Forty-eight patients were enrolled in the study (43 with AML, 3 with ALL, and 2 with MDS) and 10 cohorts constituted the trial with 3 of 6 patients involved in the tenth cohort (134 units/m^2^) who experienced DLTs. Adverse events (AE) appeared dose dependent, and augmented doses of drug were associated with a significantly higher rate of grades 3 and 4 AE. Two patients (9%) experienced clinical congestive heart failure (CHF) and 3 patients (13%) had a >10% loss in ejection fraction without overt clinical toxicity among 23 patients having a baseline cardiac function test. Patients who developed cardiac AE had received a high anthracycline cumulative dose.

The study was amended to reduce previous anthracycline exposure to 500 mg/m^2^. A hypertensive crisis (1 patient), CHF (1 patient), and cytopenias beyond 8 weeks (1 patient) were the 3 DLTs in this trial; all occurred at the doses of 134 unit/m^2^. The MTD was identified at 101 unit/m^2^ that was the dose established for the subsequent phase II clinical trials. Focusing specifically on outcomes of AML cases, 31 (72%) of 43 patients had been treated with a prior standard “3+7” regimen, and 8 of them achieved a CR after receiving CPX-351. In total, 10 of 43 patients (23%) achieved a CR and/or complete remission with incomplete count recovery (CRi). Overall, CR rate was 19% in patients aged 60 years or older and 29% in patients <60 years. The median duration of response was 6.9 months with 3 patients with ongoing remission of >1 year after study completion.

### 4.2. Phase II and III Trials

A phase II clinical trial (NCT00788892) included older patients (60–75 years) with newly diagnosed AML who were randomized in a 2:1 ratio to receive CPX-351 or “3+7” in [29]. Patients who received CPX-351 (100 U/m^2^ at days 1, 3, and 5 as induction and 44 U/m^2^ as consolidation) demonstrated a higher CR/CRi rate than those treated with a “3+7” regimen (66.7% vs. 51.2%, *p* = 0.07). However, no significant differences in EFS and OS were observed when the entire population was analyzed. A subgroup analysis involving only s-AML cases showed a higher rate of CR/CRi in patients including in the CPX-351 cohort (57.6% vs. 31.6%, *p* = 0.06) who also showed a significantly better EFS (*p* = 0.08) and OS (*p* = 0.01) [29]. Count recovery after induction appeared more prolonged in the CPX-351 cohort compared with the “3+7” group, with a median of 37 vs. 28 days for platelets >100,000 and 36 vs. 32 days for ANC ≥ 1000, respectively. Grade 3–4 infection rate was higher in patients treated with CPX-351; however, no significant risk of infection-related deaths was noted (3.5% vs. 7.3%). The 30- and 60-day mortality resulted in favor of CPX-351 (3.5% vs. 7.3% and 4.7% vs. 14.6%, respectively, *p* = 0.053) [29]. Notably, 10 refractory patients who received the “3+7” regimen crossed over and were treated with salvage treatments based on CPX-351; among them, 4 patients obtained a CR/CRi (3 CR and 1 CRi) [29]. In another phase II study, patients aged 18–65 years received CPX-351 as the first salvage approach (NCT00822094). In total, 125 patients were assigned in a 2:1 ratio to receive CPX-351 (n = 81) or investigators’ choice treatment (n = 44) [30]. Patients treated with CPX-351 showed higher CR rates (CR 37% vs. 31.8% and CRi 12.3% vs. 9.1%, respectively), although the median OS did not significantly differ between the two groups (median OS 8.5 vs. 6.3 months, *p* = 0.33). The 30-day and 60-day mortality incidence appeared similar in the two cohorts; however, a lower 90-day mortality rate was documented in the CPX-351 group (21.4% vs. 37.9%). Furthermore, patients treated with CPX-351 showed a significantly prolonged delay in neutrophil (42 vs. 34 days) and platelet (45 vs. 35 days) recovery when compared with the control group, and this delay was associated with a higher incidence of infectious events, although their infection-related deaths were not significantly different from that of the control group [30].

Lancet et al. reported data on a phase III clinical trial (NCT01696084) that included older patients (60–75 years), with newly diagnosed MRC-AML or t-AML treated with CPX-351 (n = 153) and “3+7” (n = 156), respectively [29,31]. CPX-351 therapy correlated with a significantly longer OS (9.56 vs. 5.95 months; *p* = 0.003) and EFS (2.53 vs. 1.31 months; *p* = 0.021) compared with the “3+7” regimen. CPX-351 also determined better rates of CR (37.4% vs. 24.4%; *p* = 0.04) and CR/CRi (48% vs. 32.5%; *p* = 0.016). Post-hoc subgroup analyses showed significantly prolonged OS with CPX-351 in patients with AML with prior CMML or MDS, t-AML, in those who were stratified as favorable/intermediate according to cytogenetic risk classification, and those with FMS-like tyrosine kinase 3 wild-type (*FLT3*wt) [32]. Although not significant, there was a favorable trend suggesting a longer OS also in patients with unfavorable cytogenetic characteristics and in those who harbored a *FLT3* mutation treated with CPX-351. Conversely, patients with previous documented diagnosis of MDS who were previously treated with hypomethylating agents (HMAs) did not show an improved survival when receiving CPX-351 compared with “3+7” [32]. A further analysis from this phase III study suggested a potential specific benefit from CPX-351 consolidation cycles among patients who had obtained CR/CRi after receiving the CPX-351 induction approach. In fact, patients with CR/CRi after CPX-351 induction and who subsequently received 1 or 2 consolidation cycles including CPX-351 showed a significant improvement in their median OS compared with patients treated with induction and consolidation in the “3+7” cohort (25.4 vs. 8.53 months) [33]. This primary analysis provided substantial support and evidence for approval of CPX-351 by the US Food and Drug Administration and European Medicines Agency.

Recently, Lancet et al. reported data from a 5-year follow-up of the phase 3 trial [33]. Overall, the better median OS in favor of the CPX-351 vs. “3+7” regimen was maintained (9.33 vs. 5.95 months), with a higher estimated 3- and 5-year OS (21% vs. 9% and 18% vs. 8%, respectively) in the CPX-351 group. According to age at baseline, an improved median OS after receiving CPX-351 was also noted both in patients aged 60–69 years (9.59 vs. 6.87 months) and in those aged 70–75 years (8.87 vs. 5.62 months). Considering patients achieving CR or CRi, OS was superior in the CPX-351 cohort at 3 years (36% vs. 23%) and at 5 years (30% vs. 19%), and median OS was longer with CPX-351 compared with “3+7” (21.72 vs. 10.41 months). Furthermore, 56% of patients in the CPX-351 and 46% in the “3+7” arm achieving CR or CRi proceeded to hematopoietic stem cell transplant therapy (HSCT). The median OS in these subsets from the date of HSCT was not reached for CPX-351 vs. 11.65 months for the “3+7” regimen [31]. Table 1 summarizes the most significant results of the phase III trial updated after a 5-year follow-up.

These data support the hypothesis that CPX-351 has the ability to produce long-term remission and survival in older patients with newly diagnosed high-risk/secondary AML. Indeed, after a 5-year follow-up, improved OS with CPX-351 versus conventional “3+7” chemotherapy was maintained in those who underwent HSCT and among patients who achieved CR or CRi regardless of patient age. The longer OS with CPX-351 versus “3+7” in patients who underwent HSCT and in those who achieved CR or CRi suggest that potentially deeper responses may be achieved with CPX-351.

## 5. Real-Life Experiences with CPX-351

Data reported on real-life experiences based on new therapy approaches often differ from those of clinical trials [34,35], indicating that patients included in clinical trials frequently identify a non-random cohort. These issues may affect the correct interpretation and translation of results to real-life patient care. Several real-world studies including newly diagnosed AML patients treated with CPX-351 as the frontline approach tried to address these open issues and to provide more clinical data and new information on CPX-351 efficacy and toxicity [36,37,38,39].

An Italian group assessed the efficacy of CPX-351 in 71 elderly patients (median age 66 years) with s-AML who were enrolled in the Italian Named (Compassionate) Use Program [36]. The CR/CRi and PR rate was 70.4% and 8.5%, respectively; after a median follow-up of 11 months, the estimated cumulative incidence of relapse (CIR) was 20%, with a CIR significantly decreased when HSCT was performed in first CR (12 months CIR of 5% and 37.4%, respectively, for patients receiving (=20) or not (=30) HSCT). The 1-year OS was 68.6% (median not reached) with HSCT in CR1 that represented the only significant factor associated with a longer survival. At univariate analysis, the OS was not influenced by the previous HMA therapy failure, the European Leukemia Net (ELN) 2017 risk score classification, the presence of *TP53* mutation at baseline, and the MRD status after the first cycle [36].

A French multicenter study retrospectively analyzed 103 t-AML and MRC-AML treated frontline with CPX-351 [37]. The overall response rate (CR/CRi) after induction was 59% with a negative minimal residual disease (MRD) of <10^−3^ detected by flow-cytometry observed in 57% of patients who achieved CR or CRi. Patients with mutated *TP53* or *PTPN11* genes at baseline showed a significantly lower response rate in multivariate analysis. After a median follow-up of 8.6 months, the median OS was 16.1 months, with 36 patients who received HSCT showing a significantly prolonged median OS compared with non-transplanted cases. Furthermore, the authors compared results from patients <60 years treated with CPX-351 and a historical cohort of t-AML and MRC-AML patients who had received conventional chemotherapies. Notably, survival did not significantly differ between the two series of patients [37].

A German group investigated the efficacy of CPX-351 in 188 patients (29% t-AML and 71% MRC-AML) outside clinical trials (38). After induction, the CR/CRi rate was 47% with 64% of patients presenting negative MRD < 10^−3^ by flow cytometry. Of the 188 patients, 116 (62%) underwent a HSCT. After a median follow-up of 9.3 months, the median OS was 21 months and 1-year OS rate 64%. In a multivariate analysis, having a complex karyotype predicted lower rate of responses, while previous treatment with HMAs and adverse ELN2017 risk score correlated with lower OS [38].

These real-world data confirm CPX-351 as efficient treatment for high-risk AML patients showing a similar result with the phase 3 trial and facilitating HSCT in many patients with promising outcome after transplantation. Table 2 summarizes the most significant real-life experiences including the use of CPX-351.

## 6. Experiences in Pediatric Setting

The safety and pharmacokinetic profile of CPX-351 in children and young adults has been investigated in two clinical trials, which enrolled patients affected by AML or relapsed/refractory hematologic malignancies [40,41].

The CPX-MA-1201, a phase 1 trial conducted by researchers at the Cincinnati Children’s Hospital, included 27 patients aged 1–19 years with relapsed/refractory hematologic malignancies who received CPX-351 at doses of 100 Ui/m^2^ given on days 1, 3, and 5 during the induction [40]. This trial confirmed a safe profile and encouraging response rates associated with CPX-351 also in the pediatric setting. Another clinical trial, the AML1421, a Children’s Oncology Group (COG)-sponsored phase I/II study of CPX-351 for children with relapsed AML provided CPX-351 therapy in cycle 1. In order to limit toxicities with further anthracyclines, FLAG (fludarabine 30 mg/m^2^/dose on days 1–5; cytarabine 2000 mg/m^2^/dose on days 1–5; granulocyte-colony stimulating factor (G-CSF) 5 µg/kg/dose, days 1–5 and day 15 through absolute neutrophil count (ANC) > 500/µL) regimen was contemplated as cycle 2 [41]. The primary end-points of the study were to identify the recommended phase II dose (RP2D) and to evaluate the response rate following 2 cycles. In total, 38 patients (6 in the dose-finding phase and 32 in the efficacy phase) were enrolled in the study. In the dose finding phase, 1/6 patients experienced a DLT (grade 3 ejection fraction reduction) and the RP2D was defined at 135 units/m^2^ on days 1, 3, and 5. Grade ≥ 3 toxicities during the first cycle were fever/neutropenia (45%), infection (47%), and rash (40%), but no drug-related mortality was documented. The CR rate was 54%, with 85% of the patients who achieved negative MRD among responders. In 29/30 (96.7%) responders a HSCT was used as the consolidation approach [41].

Based on these data, on 31 March 2021 the FDA approved CPX-351 for pediatric patients from one year of age—AML or MRC-AML.

## 7. Treatment Toxicities

### 7.1. Adverse Events

As discussed above, death was due to AE in 17 patients (14%) of 124 deaths in the CPX-351 arm and 19 (14%) of 140 deaths in the “3+7” arm. Of the patients who died due to causes other than leukemia (54 (44%) of 124 deaths in the CPX-351 group and 66 (47%) of 140 deaths in the “3+7” group), the most common causes of death were sepsis/septic shock (six (5%) patients in the CPX-351 group and six (4%) patients in the “3+7 “group), hemorrhage or hematoma (six (5%) patients in the CPX-351 group and four (3%) patients in the ”3+7 “group) [33]. The interpretation given by the investigators was that CPX-351 produces a more prolonged neutropenia than “3+7”, and it could be responsible for an increase in infectious events, although these do not translate into a significant increase in mortality.

In the Italian real-life experience, most of the AEs were represented by infections, with fever of unknown origin (FUO) reported in 20/71 (28%), sepsis in 20/71 (28%), pneumonia in 8 patients (11.3%), and invasive fungal infections in 3 patients (4.2%). Mucositis was reported in 5 patients (7%), whereas a self-resolving diffuse skin rash was observed in 18/71 patients (25.4%). Four patients experienced alopecia (5.6%) [36].

The French study reported 12 cases of bleeding (11%) of which 6 were grade ≥ 3, 10 cases (X%) of hypertensive crisis, and 9 cases (9%) of acute heart failure. Only 4 patients presented with grade 3 gastrointestinal AEs (vomiting in 1 patient and mucositis in 3 patients). Skin rash was observed in 26 patients (25%) and alopecia in 11 patients (11%). Grade ≥ 3 AEs were reported in 101 patients (98%), including 94 patients (91%) with febrile neutropenia. In total, 37 patients (36%) had pneumonia [37].

In the German study, grade ≥ 3 AEs were reported in 130 patients (69%). They were mainly related to infectious complications (22%) and pneumonia (22%), while gastrointestinal side effects (4%) and bleeding (4%) occurred rather infrequently [38]. 

In summary, CPX-351 requires optimal management and prophylaxis to mitigate the risk of infections probably related to the more prolonged neutropenia. Mucositis and alopecia are reduced, and this translates to improved quality of life for patients [42]. One fifth of patients presents skin toxicity that is self-resolving; however, severe cases have been reported [43]. The cardiovascular toxicity, of particular interest both in the setting of pediatric and elderly patients, will require prolonged follow-up to assess delayed cardiotoxicity related to liposomal anthracycline exposure. 

### 7.2. Quality of Life

Several studies indicated that the treatment with CPX-351 has the ability to produce quality-of-life benefits compared to conventional chemotherapy [42,44,45,46]. 

An investigational analysis of a US, multi-center supportive care study in AML (NCT02975869), evaluated patient-reported outcome (PRO) measures comparing patients treated with the CPX-351 and “3+7” regimens [44]. The PRO measures were assessed at different time-points: at baseline, 2 weeks later (when studies show patients are feeling their worst during a typical induction hospitalization), and then at 1, 3, and 6 months. The PRO measures evaluated several patient experience domains: symptoms (Edmonton Symptom Assessment Scale (ESAS)), quality of life (QoL) (FACT-Leukemia and FACT-TOI), anxiety (HADS-A), depression (HADS-D), and post-traumatic stress (PTSD Checklist). At 2 weeks, CPX-351 showed benefits compared with the “3+7” regimen according to all PRO measures, especially regarding QoL (FACT-Leu: 118.02 vs. 112.56; *p* = 0.44), anxiety (HADS-A: 4.51 vs. 5.27; *p* = 0.465), and PTSD symptoms (PTSD-checklist: 27.08 vs. 28.16; *p* = 0.6). Furthermore, after 2 weeks, patients treated with CPX-351 showed lower probabilities to present worsening ESAS total symptoms (45.7% vs. 54.1; *p* = 0.172), physical symptoms (45.7% vs. 63.5%; *p* = 0.064), and clinically significant depression symptoms (27.3% vs. 37.7%; *p* = 0.159) [44]. Furthermore, although patients treated with CPX-351 had a longer index hospitalization length of stay than “3+7” (mean of 44.3 vs. 39 days; *p* = 0.072), they showed fewer hospitalizations during the 6-month follow-up period (2.82 vs. 3.55; *p* = 0.158). Furthermore, the hospitalization period after the index admission was lower for the group of patients treated with CPX-351 (17.71 vs. 22.27 days; *p* = 0.199). Patients receiving CPX-351 showed an average time of 94.08 days alive and out of the hospital, while those treated with a standard induction regimen had 91.85 days (*p* = 0.849) [44].

Cortes et al. provided a quality-adjusted time without symptoms of disease or toxicity (Q-TWiST) exploratory analysis of the phase pivotal 3 trial with the aim to compare survival quality of life between patients receiving CPX-351 vs. conventional “3+7” after 5 years of follow-up [42]. After disease progression or relapse, the Q-TWiST is a weighted parameter that determines how much of a patient’s survival time is spent with toxicities or is “useful” time [47]. Therefore, in the absence of significant quality-of-life indicators, a Q-TWiST evaluation could suggest crucial information on the value to patients of any observed survival prolongation. In this analysis, the relative Q-TWiST gain with CPX-351 versus “3+7” was 53.6% in the base case scenario and 39.8% among responding patients. Across various sensitivity analyses, the relative Q-TWiST gains for CPX-351 ranged from 48.0 to 57.6%, remaining well above the standard clinically important difference threshold of 15% for oncology [42]. 

These findings suggest that, when compared to “3+7” conventional therapy, CPX-351 treatment increases both quantity and quality of survival in older patients with newly diagnosed high-risk/secondary AML.

## 8. New Combinations and Future Directions

Novel combinations including targeted and/or other immune therapies have recently been investigated with the aim to ameliorate the efficacy outcomes of CPX-351.

In patients with R/R AML and post-HMAs failure high-risk MDS, an ongoing trial (NCT03672539) aims to investigate the safety and efficacy of CPX-351 in combination with gemtuzumab ozogamicin (GO). The induction schedule includes CPX-351 (daunorubicin 44 mg/m^2^ and cytarabine 100 mg/m^2^) administered on days 1, 3, and 5 and GO at a dose of 3 mg/m^2^ on day 1 [48]. Patients who achieve CR/CRi could receive up to 2 consolidation cycles with CPX-351 (daunorubicin 29 mg/m^2^ and cytarabine 65 mg/m^2^) on days 1 and 3 and GO at 3 mg/m^2^ on day 1. To date, 24 patients have been enrolled in the study; among them, 75% of patients previously received a bcl2 inhibitor venetoclax in combination with HMAs and/or chemotherapy. The ORR rate (CR/CRi) was 55%, but none of the patients proceeded to HSCT according to age and comorbidities. After a median follow-up of 24 months, the median OS was 5 months with a median duration of response of 7 months. Adverse events were predominantly characterized by infectious events and the 30-day mortality was 8% [48].

Consistent synergy was also reported when combining CPX-351 with FLT3 inhibitors simultaneously or with CPX-351 exposure scheduled 24 h prior to FLT3-inhibitor administration. However, pretreatment with quizartinib for 16 h has the ability to produce a population of cells (50% of the total population) that showed decreased daunorubicin fluorescence, indicating that durable FLT3 inhibition may reduce CPX-351 uptake [49]. Andews et al. reported on three s-AML cases who achieved a CR with no unusual adverse events after receiving the combination of CPX-351 and midostaurin (given at a dose of 50 mg twice a day from day 8 to day 21) [50]. The OS was 4, 12, and 17 months, respectively. Ongoing trials are investigating the use of CPX-351 with the FLT3 inhibitors (NCT04293562, NCT04128748).

CPX-351 has demonstrated promising results when used in combination with a bcl2 inhibitor venetoclax in a heavily pretreated R/R AML population, and in de novo AML. An ongoing trial has been designed with a safety lead-in phase to define a safe dose and schedule in R/R AML, followed by 2 expansion cohorts to investigate efficacy in R/R AML (Cohort A) and frontline AML (Cohort B) [51]. The initial effective dose of venetoclax was 300 mg on days 2–21 for the safety lead-in cohort, with potential drug discontinuation after day 14 in the case of bone marrow hypo-cellularity and absence of blast cells. To date, 31 patients were treated; 26 (84%) patients with R/R AML (12 patients in the lead-in phase, and 14 in the expansion cohort A) and 5 (16%) patients with frontline AML (in the expansion cohort B). Among the 5 patients with frontline AML, 4 (80%) achieved CR/CRi with 4 out of 4 (100%) responding patients who proceed to HSCT. Among the 26 R/R AML patients, 12 (46%) achieved CR/CRi with 10 out of 12 (83%) responding patients who received HSCT. The 1-year estimated OS was 75% and 39% in frontline and R/R AML, respectively, while the median relapse-free survival (RFS) was 6.7 and 10.9 months in the 2 subgroups. The 4- and 8-week mortality was 12% and 19% for the R/R cohort, all with persistent AML. No early deaths were observed in the frontline cohort [51]. 

The use of CPX-351 in combination with immune checkpoint inhibitors is also being investigated. In murine models with AML, several experiments showed that cytarabine could increase the expression of CD80 and CD86 and downregulate the expression of PD-1 [52]. It has also been documented that blast cells manipulated in vivo with cytarabine appeared more susceptible in response to cytotoxic T cell mediated killing [52]. Combining CPX-351 with immune checkpoint inhibitors could potentially augment the cytotoxic effect and induce immune surveillance, without reducing cumulative myelosuppression.

Other trials are ongoing, aimed at evaluating the efficacy and safety of CPX-351 with IDH1 and IDH2 inhibitors (NCT04493164, NCT03825796), JAK2 inhibitor ruxolitinib (NCT03878199), hedgehog pathway inhibitor glasdegib (NCT04231851), and CDK4/6 inhibitor palbociclib (NCT03844997) (Table 3).

New landscapes in CPX-351 scenarios are evaluating its potential efficacy and safety in those patients with de novo AML stratified as intermediate risk according to the ELN2017 prognostic stratification. Interestingly, comparing de novo and stringently defined secondary AMLs occurring after a documented phase of MDS, the French group identified a molecular subgroup, termed ‘secondary-type AML’, defined by mutations in either *SRSF2*, *SF3B1*, *U2AF1*, *ZRSR2*, *ASXL1*, *EZH2*, *BCOR*, and/or *STAG2* genes. Among de novo AML patients, 33.3% had secondary-type mutations [53]. It has been shown that patients older than 60 years of age harboring secondary-type AML, as defined by this 8-gene molecular signature, had inferior outcomes to those without ‘secondary-type’ mutations when treated with conventional “3+7” chemotherapy, combining cytarabine and an anthracycline (ALFA 1200 study) [53]. This was notably true among patients with ‘intermediate-risk’ disease per ELN criteria.

The incidence of ‘secondary-type’ AML mutations increases with age and with cytogenetic risk category. Approximately 50% of de novo AML patients with intermediate risk older than 50 years harbor such secondary-type mutations. New therapeutic options are necessary in patients older than 50 years with de novo AML classified as adverse risk, and in patients with intermediate risk and exhibiting secondary-type mutations. A current study (NCT052605289) will evaluate the rate of MRD negative remissions with CPX-351 used as induction and consolidation therapy compared with intensive chemotherapy in a population of non-MRC AMLs with secondary-like mutations.

## 9. CPX-351 in *TP53* Mutated AML

CPX-351 seems to exhibit potent and direct ex vivo cytotoxicity against AML blasts harboring the *FL3-ITD* mutation [54], while other genomic predictors of response to CPX-351 have been described such as *ASXL1* and *RUNX1* mutations, which were not associated with a lower ORR [37]. Despite the efficacy of CPX-351 shown in t-AML and high-risk groups [55], several data overall support the hypothesis that resistance to liposomal daunorubicin and cytarabine chemotherapy is common in AML patients with *TP53* mutations [56,57]. In a retrospective, multi-center review of patients who received at least 1 cycle of induction chemotherapy with CPX-351 at Memorial Sloan Kettering Cancer Center (MSKCC), Moffitt Cancer Center (MCC), or Weill Cornell Medical College (WCMC), CR combined with CRi were significantly higher in WT *TP53* (62%) patients compared to *TP53*-mutant patients (33%) [56]. Furthermore, responses for minimal residual disease (MRD)-negative CR also favored WT patients (36% MRD-CR WT vs. 8.3% MRD-CR *TP53* mutant) and OS also tended towards favoring WT over *TP53*-mutant patients, although this was not significantly different (*p* = 0.093) [56]. At the 61st American Society of Hematology (ASH) Annual Meeting, the Dana Farber group presented the results of a study including 309 AML patients randomized 1:1 to receive CPX-351 or “3+7” induction and consolidation to assess the impact of gene mutations on outcome. In a multivariate analysis, incorporating clinical and genetic characteristics, CPX-351 was associated with prolonged OS compared with “3+7”, but *TP53* mutations were associated with a poor prognosis, irrespective of treatment arm [57]. Specifically, median OS was longer in the CPX-351 versus “3+7” arms among patients with *DNMT3A* (12.6 vs. 5.5 months; HR = 0.41 (95% CI: 0.19–0.89)) and *TET2* (9.1 vs. 3.7 months; HR = 0.47 (95% CI: 0.23–0.93)) mutations, while median OS was similar among patients with *TP53* mutations treated with CPX-351 and “3+7” (4.5 vs. 5.1 months; HR = 1.19 (95% CI: 0.70–2.05)). Although CPX-351 combined regimens have shown promising results in patients with unfavorable prognostic risk, these treatments do not seem to improve response rates and outcomes in a TP53-mutated setting.

## 10. Conclusions and Open Questions

Lancet et al. recently reported data from a 5-year updated phase 3 trial comparing CPX-351 with conventional “3+7” chemotherapy in patients aged 60–75 years. From these results, further support of previous evidence emerged that CPX-351 may strongly contribute to long-term remission and prolonged survival [33]. The incidence of remission, a pre-requisite for long-term OS, was superior in patients receiving CPX-351, and many of those patients could proceed to HSCT. Longer OS observed with CPX-351 indicated deeper responses to this agent in the high-risk AML population. These promising results have also been confirmed by several real-life experiences, which demonstrated the efficacy of CPX-351 outside of the clinical trials setting and validated data on the safer toxicity profile of this agent compared with standard chemotherapy.

Despite these attractive results, some questions remain open. First, it is not yet clear which subset of patients will benefit from CPX-351. Several studies reported promising outcome results in patients identified as high-risk, regardless of their age and AML subtypes. However, since CPX-351 has been shown to produce a higher remission rate in this setting, longer duration of response and improved OS in favorable or intermediate-risk AML cannot be excluded via a better impact on leukemia-initiating cells associated with the CPX-351 specific bone marrow tropism.

Another unresolved issue is how best to use CPX-351. Should CPX-351 be contemplated only as an induction approach or should it be considered as part of consolidation or maintenance, and as a bridge to HSCT? When a HSCT is considered, is consolidation treatment with CPX-351 useful, or is this option required only to maintain a satisfactory response until a donor becomes available or to induce a deeper response that allows patients to receive an allogenic transplant with less possible leukemia burden? The persistence/reappearance of MRD represents one of the most important unfavorable prognostic factors in an AML treatment scenario. The significant efficacy of CPX-351 on leukemia-initiating cells suggests that CPX-351 could play a crucial role in the disappearance of persistent MRD and potentially be employed in case of relapse after HSCT. Moreover, CPX-351 may represent the chemotherapy backbone to be associated with novel agents, as delineated in Section 8.

Yet another unanswered issue is the potential onset of resistance in patients treated with CPX-351, especially in those patients who have not received anthracycline and could have a greater probability of developing biological resistance and clinical cardiac toxicities.

CPX-351 in newly diagnosed AML-MRC and t-AML patients aged 60–75 years has resulted in superior remission rates compared to conventional chemotherapy, improvements in EFS and OS, and an increased uptake of HSCT. Future directions include a complete evaluation of dose intensification with CPX-351 in high-risk settings, combining this agent with targeted therapies, a better mechanistic understanding of improved responses in t-AML and AML-MRC, and an assessment of its drug efficacy and toxicity in low- and intermediate-risk settings.

## Figures and Tables

**Table 1 cancers-14-02843-t001:** Results from the 5-year phase III trial comparing efficacy of CPX-351 vs. “7+37+3” in high-risk AML patients (reference number 33).

Treatment		Median OS ^1^ (Range)		OS ^1^ Rate at 3 Years (%)		OS ^1^ Rate at 5 Years (%)	
**CPX-351**	**3+7**	**CPX-351**	**3+7**	CPX-351	3+7	CPX-351	3+7
All patients		9.33 months (6.37–11.86)	59.93 months (59.73–60.50)	21	9	18	8
N = 153	N = 156						
Patients aged 60–69 years		9.59 months (6.01–12.62)	6.87 months (4.63–8.84)	23	14	20	0
n = 96	n = 102						
Patients aged 70–75 years		8.87 months (4.73–12.19)	5.62 months (3.29–7.52)	18	12	16	0
n = 57	n = 54						
Patents who achieved CR ^2^/CRi ^3^		21.72 months (13.01–29.70)	10.41 months (7.82–15.21)	36	23	30	19
n = 73	n = 52						
Patients who received a HSCT ^4^		NR ^6^	11.65 months (4.57–24.28)	58	29	/	/
n = 41	n = 24						
Patients with previous HMA ^5^ exposure who had CR ^2^ or CRi ^3^		14.72 months (7.75–55.56)	10.17 months (4.86–17.91)	/	/	/	/
n = 23	n = 20						
Patients with previous HMA ^5^ exposure who had CR ^2^ or CRi ^3^ and proceed to HSCT ^4^		NR ^6^	14.09 months (2.14–not estimable)	/	/	/	/
n = 13	n = 7						
Responder patients who relapsed		3.16 months, (9.33–16.82)	7.82 months, (4.86–13.40)	/	/	/	/
n = 22	n = 15						

^1^ OS = overall survival. ^2^ CR = complete remission. ^3^ CRi = complete remission with incomplete count recovery. ^4^ HSCT = hematopoietic stem cell transplantation. ^5^ HMA = hypomethylating agents. ^6^ NR = not reached.

**Table 2 cancers-14-02843-t002:** Real-life experiences with CPX-351.

References	Number of pts ^1^/Median Age	Median Age (Years)	Overall Response Rate (CR ^2^/Cri ^3^) after Induction (%)	MRD ^4^ (10<3) Negativity Rate (%) in Evaluable pts ^1^	Cumulative Incidence of Relapse (%)	Median Follow-Up	Median OS ^5^	Pts Receiving HSCT ^6^ after Response (%)	Negative Prognostic Factors on OS ^5^	Negative Prognostic Factors on OS ^5^
Italian group	71 (36 s-AML, 22 t-AML, 13 MRC-AML)	66	70.4	37.5	23.6	12 months	1-year OS ^5^ 68.6%	20	Complex karyotype	HSCT ^6^ performed in first CR ^2^
French group	103 (27 t-AML, 74 MRC-AML, 2 other)	67	59	57	25	8.6 months	16.1 months	37	Monosomal karyotype, *DNMT3A* mutation, *TP53* mutation	Presence of spliceosome mutations
German group	188 (56 t-AML, 132 MRC-AML)	65	47	64	23 after transplant	9.3 months	21 months; 1-year OS ^5^ 64%	62	Pretreatment with HMA ^7^, adverse ELN2017 risk, complex karyotype, MRD positivity after induction not undergoing allo-HSCT ^6^	Intermediate ELN2017 risk, no pretreatment with HMA ^7^
UK group	57 (8 t-AML, 29 s-AML, 11 MRC-AML, 9 other)	63	61	/	/	12 months	429 days	38	*TP53* mutation	*Wt-TP53* mutation

^1^ pts = patients. ^2^ CR = complete remission. ^3^ Cri = complete remission with incomplete count recovery. ^4^ MRD = minimal residual disease. ^5^ OS = overall survival. ^6^ HSCT = hematopoietic stem cell transplantation. ^7^ HMA = hypomethylating agents.

**Table 3 cancers-14-02843-t003:** Ongoing trials including CPX-351 in combination with other agents.

Disease	Characteristic	Combination	NCT Number	Phase
R/R ^1^ AML ^2^	CD33 positive (>3%). Excluding prior treatment with CPX-351 or GO ^6^	GO ^6^ -induction: GO ^6^ on day 1 -consolidation: GO ^6^ on day 1 -maintenance: GO ^6^ on day 1 every 6 weeks for up to 6 cycles	NCT03672539	Phase 1
Frontline AML ^2^	Age > 55 years. Excluding prior treatment with CPX-351, HSCT ^5^ or GO ^6^	GO ^6^ Induction: -cohort A: GO ^6^ on day 1 -cohort B: GO ^6^ on days 1, 4 -cohort C: GO ^6^ on days 1, 4, 7 Consolidation: -GO ^6^ on day 1	NCT03878927	Phase 1
Frontline, post-MPN ^3^ AML ^2^	Secondary AML transformed from MPNs ^3^ (PV ^7^, ET ^8^ and MF ^9^)	Ruxolitinib -identification of the maximum-tolerated dose of ruxolitinib in combination with CPX-351	NCT03878199	Phase 1/2
R/R ^1^ AML ^2^	IDH2 mutated. Not excluding prior IDH2 inhibitor treatment	Enasidenib -identification of the maximum-tolerated dose of enasidenib in combination with CPX-351	NCT03825796	Phase 2
Frontline and R/R ^1^ AML ^2^ and HR-MDS ^4^	IDH1-R132 mutated. Excluding patients with prior anthracycline exposure of >360 mg/mq daunorubicin	Ivosidenib -induction: 500 mg/die on days 1–28 -consolidation: 500 mg/die on days 1–28 -maintenance: 500 mg/die for up to 2 years in the absence of disease progression or unacceptable toxicity	NCT04493164	Phase 2
Frontline AML ^2^	Excluding prior treatment with cell cycle inhibitors or CPX-351	Palbociclib Induction: - palbociclib administered orally on days -1 and -2 at 125 mg PO during the phase IIa portion (dose level 0). Day 0 will be rest and then Palbociclib on days 2, 4, and 6 followed by rest/monitoring period (days 7–28)	NCT03844997	Phase 1/2
Frontline and R/R ^1^ AML ^2^	Excluding prior treatment with CPX-351 or venetoclax	Venetoclax -induction: venetoclax 300 mg/die on days 2–21 -consolidation: venetoclax 300 mg/die on days 2–21	NCT03629171	Phase 2
R/R ^1^ Acute Leukemia	Ages 1–39 years *AML/T-ALL/ETP-ALL/MPAL/AUL/KMT2A* rearranged ALL ^10^	Venetoclax induction and consolidation: -dose level 0–400 mg daily for 21 days -dose level -1–400 mg daily for 14 days	NCT03826992	Phase 1/2
R/R ^1^ AML ^2^	FLT3 mutated. Including patients who received a prior FLT3 inhibitor	Quizrtinib -induction: 30 mg on days 8–21 -consolidation: 30 mg on days 6–21 -maintenance 30 mg daily	NCT04209725	Phase 2
Frontline AML ^2^	Previously untreated t-AML or MRC-AML. Excluding patients who received prior CPX-351 or glasdegib	Glasdegib -induction: 100 mg daily on days 6 to 28 -consolidation: 100 mg daily on days 6 to 28 -maintenance: 100 mg daily for up to one year	NCT04231851	Phase 2

^1^ R/R = resistant/relapsed. ^2^ AML = acute myeloid leukemia. ^3^ MPN = myeloproliferative neoplasms. ^4^ HR-MD = high-risk myelodysplastic syndromes. ^5^ HSCT = hematopoietic stem cell transplantation. ^6^ GO = gemtuzumab ozogamicin. ^7^ PV = polycythemia vera. ^8^ ET = essential thrombocytosis. ^9^ MF = myelofibrosis. ^10^ ALL= acute lymphoblastic leukemia.

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
