# Peer review of "CPX-351: An Old Scheme with a New Formulation in the Treatment of High-Risk AML"

_cancers, 2022, doi:10.3390/cancers14122843_

Round 1

Reviewer 1 Report

The authors present an exhaustive review relative to the effects of CPX-351 administration in high-risk AML patients, reporting in detail the results of the main trails and clinical experiences.

Overall, the work is well structured, including tables well summarizing the content. The bibliography used is mostly updated and appropriate, apart some missing items (as reported below).

The paragraphs are organized in a proper way, even if the content is not always clear and well described and contains some inaccuracies, as pointed below:

  • line 57-67: better explain the three subgroups classification, listing them: t-AML is part of s-AML?
  • Phase I clinical study description:
    • the cited papers are dated 2011 and 2012; I don’t think that the first trial initiated in 2015.
    • add some considerations on the conclusions of phase I clinical trials, especially on the fact that CPX-351 at high doses is suitable for patients requiring intensive chemotherapy and at reduced doses in those with advanced age and/or co-morbidities
  • Phase II clinical study: the conventional treating regimen is named 7+3 or 3+7?
  • Table 1: the total number of patients in the 7+3 group is 156 according to reference 33
  • There is no uniformity in the writing of numbers along the text (line 243 for example).
  • Conclusion section: I would add some considerations on the new promising combination of CPX-351 with other molecules that are currently under study. I would especially speculate on how they can help in answering to all the open questions correctly reported by the authors.

English requires adjustments. The sentences are often too long and redundant and many typos and small mistakes are present in the text. Here just some examples:

Line 27: please correct with “no significant changes in overall survival have been observed”

Line 34: please remove “antecedent”

Line 46: “is hypothesized to determine” please replace with “drives”

Line 56: “According to a biological profile” replace with “From a biological perspective”

Line 57: better to use “that can be classified in…”

Line 92-93: please rephrase

Line 112: “characterized” instead of “characterizing”

Line 120: “occurring” in not the proper term here

Line 129: replace “the majority of which are included” with “mainly located”

Line 160: remove “attack DNA and”

Line 167-171: please rephrase

Line 178-181: the meaning of this sentence is not clear to me

Line 184: remove “aimed”

Line 184-189: split in two sentences

Line 303: “included” instead of “including”

Line 373: remove “a”

Line 375: better to say “from one year of age”

Line 378-379: please rephrase

Line 389:  define acronym FUO

Line 461: remove “was”

Line 493-494: I do not find the reported information in the mentioned reference

Line 493: reference 56 is not present in the bibliography

Line 518: reference 57 is not present in the bibliography

Author Response

  1. line 57-67: we better explain the three AML subroups; s-AML including t-AML
  2. in the phase 1 study description the year has been changed
  3. The MTD established was 101 unit/mq and there was no difference acording to age and comorbidities
  4. 7+3 was modified in 3+7
  5. table 1 was modified
  6. the numbers in the text was modified
  7. there is an entire parghraph on novel combination approaches based on CPX-351 treatment including detailed data and potential impact on AML therapy scenario
  8. All the revisions in the text have been modified

Reviewer 2 Report

This is an interesting review with clinical relevance. It might be improved by a more detailed discussion of CPX-351's potential in molecularly defined s/t-AML subpopulations.

For instance, a recent Blood paper by Tim Grob et al. has recently suggested that TP53 mutated MDSs and AMLs should be considered a separate entity with a particularly poor outcome (2y OS about 12%). As TP53 mutations are the most frequent in t-AML, this issue should be discussed in detail along with data about CPX-351 vs other therapies in this subset.

In general, as AML classification is more and more linked to specific mutational signatures, it might be ineteresting to focus on what of them are associated to better or worse clinical outcomes in s/t-AMLs.  

Several sentences are not supported by references, e.g.: line 533: Lancet et al... lacks reference number.  

Author Response

  1. we have introduced an entire paragraph on the impact of CPX-351 on TP53 mutated AML patients
  2. we also added some sentences on the impact of CPX-351 in AML patients according to specific mutations
  3. all the sentences were supported by specific references

Round 2

Reviewer 1 Report

There are still minor typos in the text but the work can be accepted in this form.

Reviewer 2 Report

The manuscript has been significantly improved, I have no further suggestions.